# Cardiac activity impacts cortical motor excitability

Esra Al[1,2,3,4,5]*, Tilman Stephani[1,6], Melina Engelhardt[7,8], Saskia Haegens[4,5,9], Arno Villringer[1,2,3], Vadim V. Nikulin[1]

1 Department of Neurology, Max Planck Institute for Human Cognitive and Brain Sciences, Leipzig, Germany, 2 MindBrainBody Institute, Berlin School of Mind and Brain, Humboldt-Universität zu Berlin, Berlin, Germany, 3 Center for Stroke Research Berlin (CSB), Charité–Universitätsmedizin Berlin, Berlin, Germany, 4 Department of Psychiatry, Columbia University, New York, New York, United States of America, 5 Division of Systems Neuroscience, New York State Psychiatric Institute, New York, New York, United States of America, 6 International Max Planck Research School NeuroCom, Leipzig, Germany, 7 Charité–Universitätsmedizin Berlin, Klinik für Neurochirurgie, Berlin, Germany, 8 Charité–Universitätsmedizin Berlin, Einstein Center for Neurosciences, Berlin, Germany, 9 Donders Institute for Brain, Cognition and Behaviour, Radboud University, Nijmegen, the Netherlands

* esraal@cbs.mpg.de

**Data Availability Statement:** The experimental code and analysis scripts are available at https://github.com/Esra-Al/Cardiac_Motor_TMS_EEG. Access to the data and code for all analytical figures can be obtained at the following link:

## Abstract

Human cognition and action can be influenced by internal bodily processes such as heartbeats. For instance, somatosensory perception is impaired both during the systolic phase of the cardiac cycle and when heartbeats evoke stronger cortical responses. Here, we test whether these cardiac effects originate from overall changes in cortical excitability. Cortical and corticospinal excitability were assessed using electroencephalographic and electromyographic responses to transcranial magnetic stimulation while concurrently monitoring cardiac activity with electrocardiography. Cortical and corticospinal excitability were found to be highest during systole and following stronger neural responses to heartbeats. Furthermore, in a motor task, hand–muscle activity and the associated desynchronization of sensorimotor oscillations were stronger during systole. These results suggest that systolic cardiac signals have a facilitatory effect on motor excitability—in contrast to sensory attenuation that was previously reported for somatosensory perception. Thus, it is possible that distinct time windows exist across the cardiac cycle, optimizing either perception or action.

## Introduction

How we perceive and engage with the world is influenced by the dynamic relationship between the brain and the rest of the body including respiratory, digestive, and cardiac systems [1–6]. For example, cardiac activity has been found to influence visual and auditory perception [7–9]. In the domain of somatosensation and pain, perception and neural processing of stimuli have been reported to decrease during the systolic compared to the diastolic phase of the cardiac cycle [10–13]. An overall systolic dampening of cortical processes was suggested to be due to baroreceptor activation during systole [14]. In support of this view, reaction times to auditory, visual, and tactile stimuli have been shown to be slower for systolic presentation [15,16].

**Funding:** For this project, EA received funding from the Max Planck School of Cognition, Walter Benjamin Program of the German Research Foundation (DFG) and the experiment was funded by the Max Planck Society. The funders had no role in study design, data collection and analysis, decision to publish, or preparation of the manuscript.

**Competing interests:** The authors have declared that no competing interests exist.

**Abbreviations:** BEM, boundary element model; ECG, electrocardiography; EEG, electroencephalography; EMG, electromyography; FDI, first dorsal interosseous; HEP, heartbeat-evoked potential; MEP, motor-evoked potential; MVC, maximal voluntary contraction; TEP, TMS-evoked potential; TKE, Teager–Kaiser Energy; TMS, transcranial magnetic stimulation.

Furthermore, when touch was initiated during systole, the duration of active sensation was longer [17]. However, for some other motor activities, a facilitatory effect of systole has been observed, for instance, self-initiated movements occurred more frequently during systole [17,18] or around the R-peak [19]; but see [20]. Similarly, an effect of the cardiac cycle was also observed in gun shooting, with shooters preferentially triggering a gun in a cardiac window that included a large part of systole and the initial part of diastole [21]; but also see an opposite effect for elite shooters [22]. Furthermore, (micro)saccades occur more often during systole whereas fixations during diastole [23,24]. One possible explanation for these effects can be related to blood movement within the body (peripheral), rather than being driven by top-down neural mechanisms (central) [23]. In addition to cardiac phase effects, stronger neural responses to heartbeats, i.e., heartbeat-evoked potentials (HEPs), are followed by increases in visual perception and decreases in somatosensory detection [11,13,25]. These results, therefore, suggest that cardiac activity could interact with both perception and action. However, it is not known whether a central/cortical mechanism underlies these cardiac effects.

One possibility is that cardiac activity exerts its effects through alterations of neuronal excitability in different parts of the brain. A few previous studies have investigated this hypothesis using transcranial magnetic stimulation (TMS) over the primary motor cortex, which induces motor-evoked potentials (MEPs), an indicator of corticospinal excitability. However, so far, no main effect of the cardiac phase on excitability levels has been observed [26–29]. Otsuru and colleagues [28] showed corticospinal excitability 400 ms after R-peak to be significantly higher in poor compared to good interoceptive perceivers, but the authors did not observe a significant main effect of cardiac phase on motor excitability. There are several possible methodological reasons for these findings. Importantly, the examination of excitability was limited to specific time intervals (only up to 400 ms after R-peak), rather than across the entire cardiac cycle including both systolic and diastolic window. Furthermore, individual brain anatomy was not taken into consideration, possibly resulting in higher variability of the stimulated brain regions. Lastly, these previous studies only included peripheral measures of corticospinal excitability without direct measurement of cortical excitability via concurrent electroencephalographic (EEG) recordings. This approach requires considering multiple concomitant factors to minimize TMS artifacts (see a recent review [30]). To address this limitation, stimulation intensity needs to be minimized by carefully determining the motor threshold, the effects of TMS clicks should be minimized to avoid acoustic stimulation, the experiment should have a sham condition to account for bone-conducted sound from the coil vibration [31] and thorough data cleaning to remove TMS artifacts using spatial (for instance, ICA) and temporal filters.

In the present study, we systemically examine whether cortical and corticospinal excitability change across the entire cardiac cycle and whether they interact with HEPs. Additionally, to comprehensively evaluate cardiac-related activity, we investigated variations in heart rate following TMS stimulation throughout the cardiac cycle. Neuronavigated TMS was used in combination with multichannel EEG, in order to comprehensively investigate both cortical and peripheral TMS-evoked responses. If systolic activity attenuates motor excitability, similar to somatosensory perception, TMS pulses during systole would be expected to produce weaker MEPs in the hand muscle and weaker TMS-evoked potentials (TEPs) in the motor cortex. Similarly, increases in HEPs would be expected to attenuate upcoming motor excitability. Alternatively, if behavioral observations of motor facilitation during systole are correct [18,21,23,24], stronger MEPs and TEPs should be observed. Supporting this latter hypothesis, we found that both peripheral and central TEPs were in fact higher during systole. Moreover, stronger HEPs preceded increases in excitability. In line with these findings, hand–muscle activity and associated desynchronization of sensorimotor oscillations in a motor pinch task

were strongest during the systolic heart phase. In addition, in an exploratory analysis, we found a specific influence of TMS on heart rate depending on the cardiac phase, which could be relevant for clinical studies. Taken together, our results reveal that there is a facilitatory effect of systolic activity on motor excitability, possibly connected with an optimal window for action initiation during the cardiac cycle.

## Results

### Motor-evoked potentials change across the cardiac cycle

We first aimed to determine whether corticospinal excitability changes systematically over the phases of the cardiac cycle. For this purpose, we applied TMS to the right primary motor area at random time points across the cardiac cycle and recorded MEPs of the first dorsal interosseus muscle in the left hand in 36 participants. Following the TMS administration, the corresponding cardiac phase was determined posteriori (see Methods for the details; Fig 1). Consistent with the notion of corticospinal excitability changes across the cardiac cycle, MEP amplitudes were significantly higher during systole to diastole (Wilcoxon signed rank test, $V = 463$, $p = 0.040$, Cohen's d = 0.341; Fig 2). As an additional control of a potential effect of ECG artefact on EMG activity, systolic and diastolic EMG activity during the sham condition was subtracted from real TMS activity. After this control analysis, MEP amplitudes remained significantly higher during systole ($V = 463$, $p = 0.041$, Cohen's d = 0.340). So, these results suggest that corticospinal excitability is higher during systole compared to diastole.

### TMS-evoked potentials vary across the cardiac cycle

In addition to corticospinal excitability, we also aimed to test differences of purely cortical responses to the TMS pulse between the systole and diastole phases of the cardiac cycle. Cortical excitability was probed by early TEPs (15 to 60 ms post-TMS) measured from a cluster of electrodes (C4, CP4, C6, CP6) over the right motor cortex (hotspot). By focusing on this time window, we prevented a possible contamination of cranial muscle artefacts in the very early peaks as well as sensory-evoked potentials in the later time windows [30]. TEP amplitudes between 22 and 60 ms following the TMS stimulation were stronger during systole as compared to diastole (cluster-based permutation t test, $p_{cluster} = 0.009$; Cohen's d = 0.41; Fig 3A). To test whether these results were indeed related to neural activity of the cortex, rather than reflecting TMS and cardiac artifacts, we contrasted them with the sham TMS condition. During the sham condition, a plastic block between the coil and the participant's head was placed to keep air- and bone-conducted auditory and somatosensory sensations similar to the real TMS [31]. Additionally, pulse-related artifacts in the EEG were expected to be similar to the real TMS stimulation. Therefore, if the cardiac cycle effect on TEPs is a genuine modulation of evoked neuronal responses to TMS, it should not be present in the sham condition. A cluster-based permutation test did not reveal any significant difference in TEPs in response to sham TMS during systole and diastole ($p_{cluster} = 0.2$; Fig 3B). Furthermore, to account for physiological and stimulation artifacts, TEPs during sham were subtracted from those in the real TMS condition. The TEP difference was significantly higher between 24 and 60 ms during systole relative to diastole ($p_{cluster} = 0.008$, Cohen's d = 0.41; Fig 3C). The corresponding neural sources of the TEP difference between systole and diastole were observed to be maximal around the right primary motor cortex (Fig 3D). These results show that similar to corticospinal excitability, purely cortical excitability increases during systole, suggesting increased neuronal excitability in the motor system.

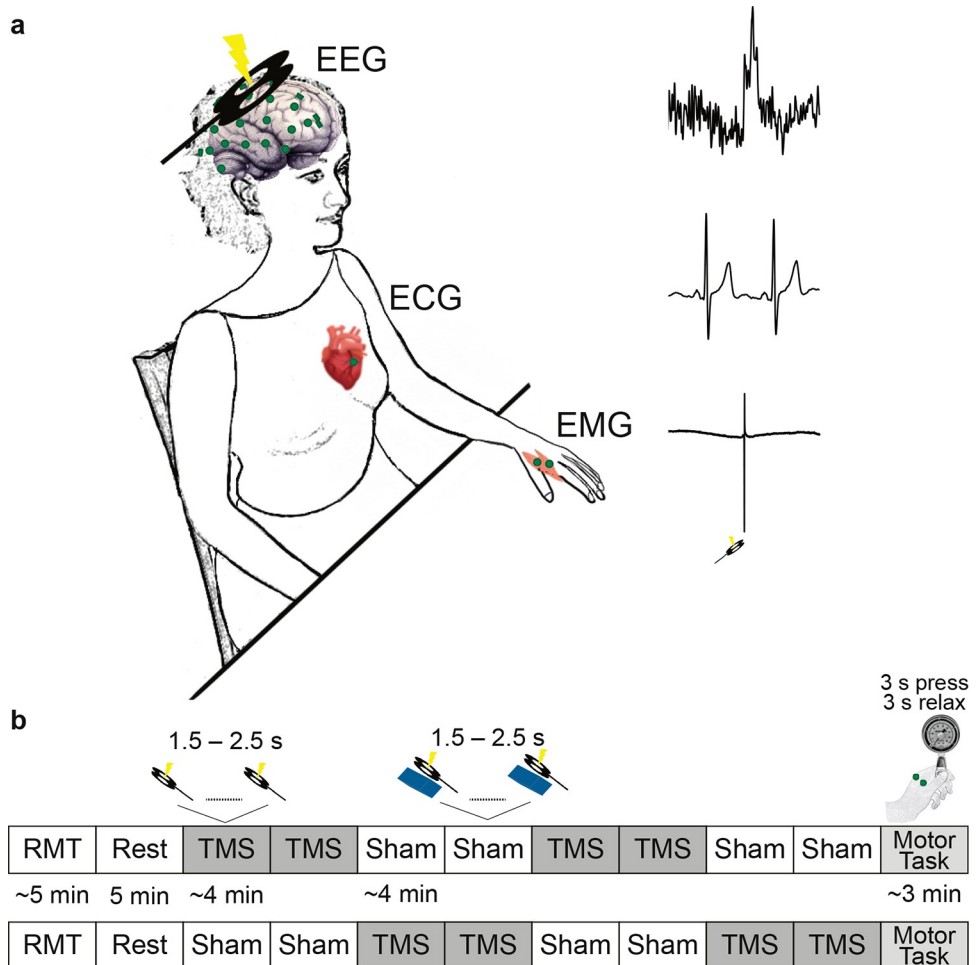

**Fig 1. Experimental paradigm.** (**a**) TMS was applied over the right primary motor cortex of the participants. The motor response to TMS in their left hand, i.e., MEP, was measured by bipolar EMG. Their cortical responses to TMS, the TEP, as well as to heartbeats, the HEP, were measured using multichannel EEG. The heart activity was recorded via ECG. (**b**) After determining the individual RMT, participants underwent a resting-state EEG measurement. Thereafter, 416 single TMS pulses with an intensity of 120% of the RMT were applied in 4 blocks. There were also 4 blocks of sham conditions, in which a plastic block was placed between the TMS coil and the head of the participant. The pairs of real and sham TMS blocks were randomized across the participants. At the end of the TMS blocks, participants performed a motor pinch task. In this task, they were instructed to squeeze a pinch gauge with their left thumb against the index finger while a red circle was presented in the middle of the monitor. When the circle became green, they relaxed their fingers. In this order, participants performed 30 trials. ECG, electrocardiography; EEG, electroencephalography; EMG, electromyography; HEP, heartbeat-evoked potential; MEP, motor-evoked potential; RMT, resting motor threshold; TEP, TMS-evoked potential; TMS, transcranial magnetic .stimulation

## Muscle-related peripheral and central activity fluctuates across the cardiac cycle

To determine whether changes in motor excitability during systole facilitate muscle activity in the absence of TMS stimulation, we analyzed cardiac effects on muscle activity in a follow-up motor pinch task. During the task, participants were asked to pinch a dynamometer with their index finger and thumb, while we recorded EMG, EEG, and ECG signals (**Fig 4A**). To estimate peripheral muscle force, we calculated the envelope of EMG activity when participants initiated the pinch during systole and diastole (see EMG envelope in the Methods section). Cluster statistics revealed a significant increase in the EMG envelope from 340 to 454 ms after the

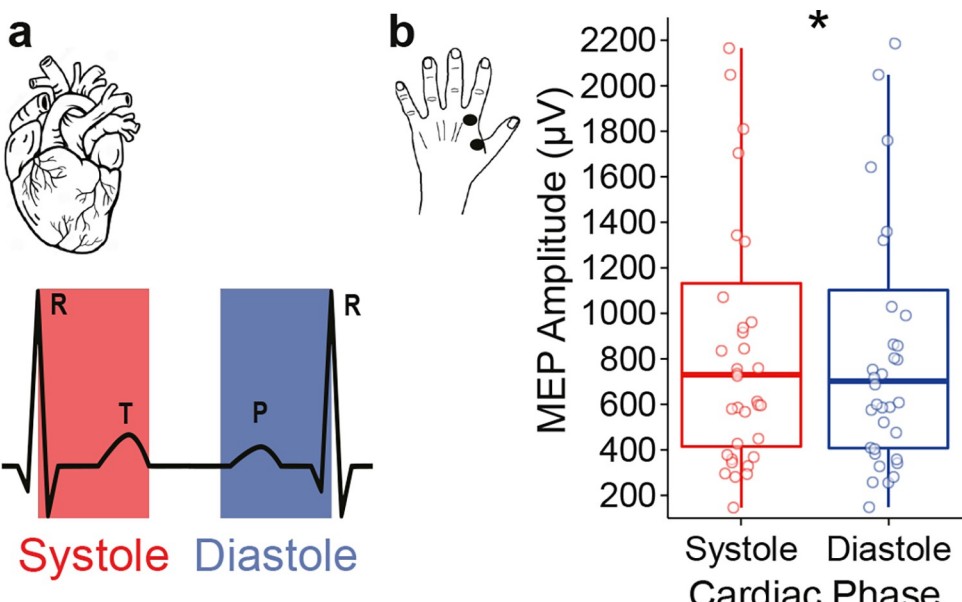

**Fig 2. Changes in corticospinal excitability depending on the timing of TMS application across the cardiac cycle.** (**a**) Schematic of the cardiac cycle. The systolic phase (indicated in red) starts with the R-peak and reflects the ventricular contraction of the heart (leading to blood ejection), whereas the diastolic phase (indicated in blue) represents the relaxation phase during which the heart refills with blood. To equate the probability of a stimulation event occurring in either of the 2 phases, equal lengths of systole and diastole were defined. (**b**) MEP amplitudes of the first dorsal interosseus muscle in the left hand are higher in response to TMS stimulation during systole (red) compared to diastole (blue). The individual points indicate the mean MEP amplitudes for each participant. *$p < 0.05$. Access to the data and code for all analytical figures can be obtained at the following link: https://osf.io/eg8dz?view_only=7edb12b6e50e4c709ad60579bf5ebd62.

onset of the pinch during systole compared to diastole ($p_{cluster} = 0.02$, Cohen's d = 0.44; **Fig 4A**). To test whether this finding might have been related to blood circulation–related changes in the fingers, we sampled systolic and diastolic EMG activity during the resting state condition. This analysis did not reveal any significant difference in the resting EMG envelope between systole and diastole (no significant clusters were found). This indicates that there was no influence of cardiac-related artifacts on the EMG signal across the cardiac cycle.

Following the analysis of the muscle activity in the periphery, we also tested whether sensorimotor oscillations in the motor areas desynchronize differently following the initiation of the pinch during systole and diastole. This analysis demonstrated that the desynchronization of sensorimotor oscillations in the range of 8 to 25 Hz was stronger between 0 and 726 ms following pinch onset during systole as compared to diastole ($p_{cluster} = 0.012$, Cohen's d = 0.37; **Fig 4B and 4C**). To investigate whether this finding was influenced by cardiac-related artifacts, we again sampled systolic and diastolic windows during the resting state and tested the differences in sensorimotor oscillations between systole and diastole. Also, in this control analysis, no significant differences were found ($p_{cluster} = 0.12$). Thus, these results indicate that both peripheral muscle activity and its central correlates are stronger when the movement starts during systole as compared to diastole.

## Heart rate changes depending on the timing of TMS across the cardiac cycle

Previous research suggests that TMS stimulation has significant effects on heart rate [32]. However, it is not known whether this effect depends on the cardiac phase of the TMS

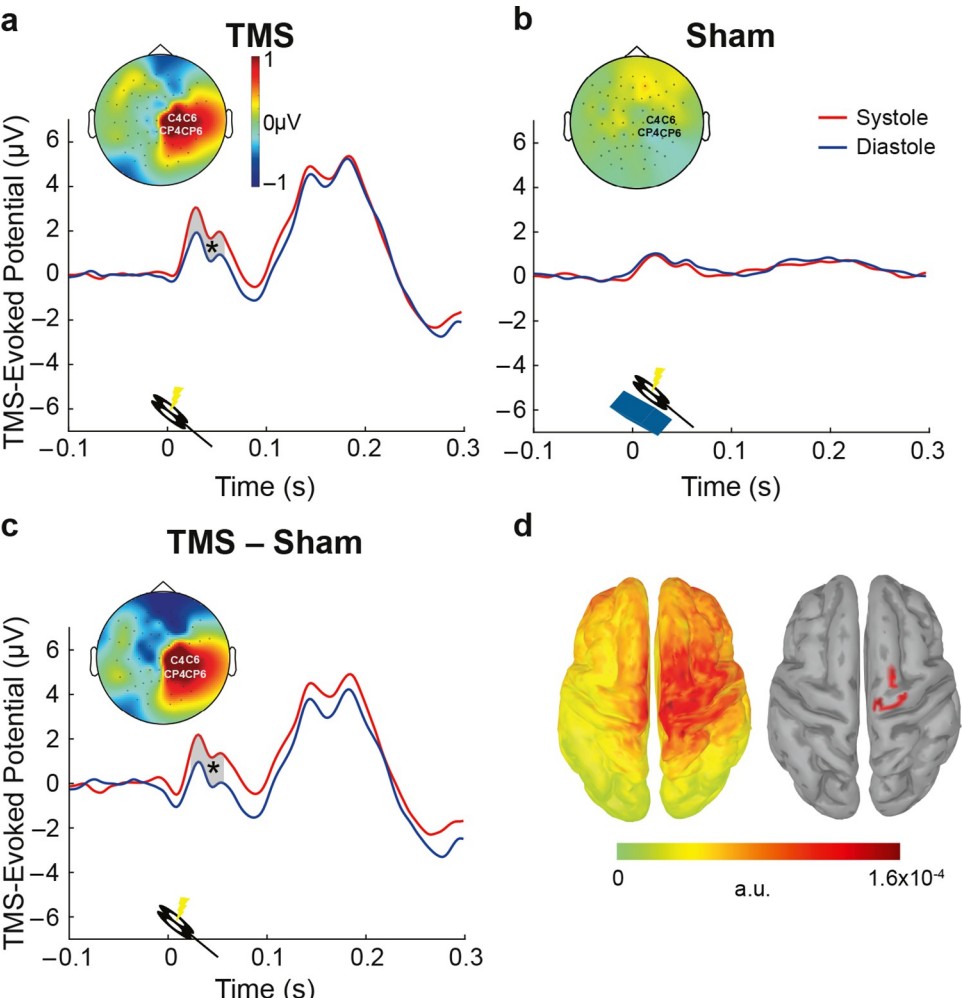

**Fig 3. Changes in cortical excitability across the cardiac cycle.** (**a**) TEPs, in response to TMS stimulation, at the electrodes closest to the motor hotspot (C4; CP4; C6; CP6). Early TEPs were significantly larger during systole compared to diastole in motor areas between 22 and 60 ms. The contrast between systole and diastole in this time window is shown in the topography plot. (**b**) Same as in (**a**), for the sham TMS condition. No significant differences between systole and diastole were observed here. (**c**) The difference curve between real TMS and sham, for systole and diastole (sham-corrected TEP contrast). After correcting for the TMS and physiological artifacts, TEPs during systole and diastole were significantly different between 24 and 60 ms. (**d**) The source reconstruction of the corrected TEP contrast (systole minus diastole) between 24 and 60 ms (left), and same displaying the strongest generators only (thresholded at 85% of the maximum activity and clusters sizes of at least 5 vertices; right).

stimulation. Therefore, in an exploratory analysis, we investigated the changes of the heart rate in response to TMS stimulation during systole and diastole. As a measure of heart rate, we calculated the length of interbeat intervals in the cardiac cycles before TMS (pre-TMS), during TMS, and post-TMS stimulation. The analysis showed a main effect of time ($F_{2, 70} = 23.11$, $p = 2 \cdot 10^{-8}$, $ges = 2 \cdot 10^{-4}$) and an interaction of time and cardiac phase ($F_{2, 70} = 10.30$, $p = 1 \cdot 10^{-4}$, $ges = 2 \cdot 10^{-4}$) on heart rate. Comparison of heartbeat intervals preceding TMS and concurrent with TMS revealed a significant cardiac deceleration when TMS stimulation occurred during systole ($t_{35} = -5.73$, $p = 2 \cdot 10^{-6}$, Cohen's d = 0.96). This was followed by a cardiac acceleration (from TMS to post-TMS; $t_{35} = 8.58$, $p = 4 \cdot 10^{-10}$, Cohen's d = 1.43; **Fig 5**). No significant changes were observed for stimulations during diastole (from pre-TMS to TMS, $t_{35} = 0.75$, $p = 0.5$ and from TMS to post-TMS, $t_{35} = 0.42$, $p = 0.68$; **Fig 5**). Post hoc $t$ tests showed

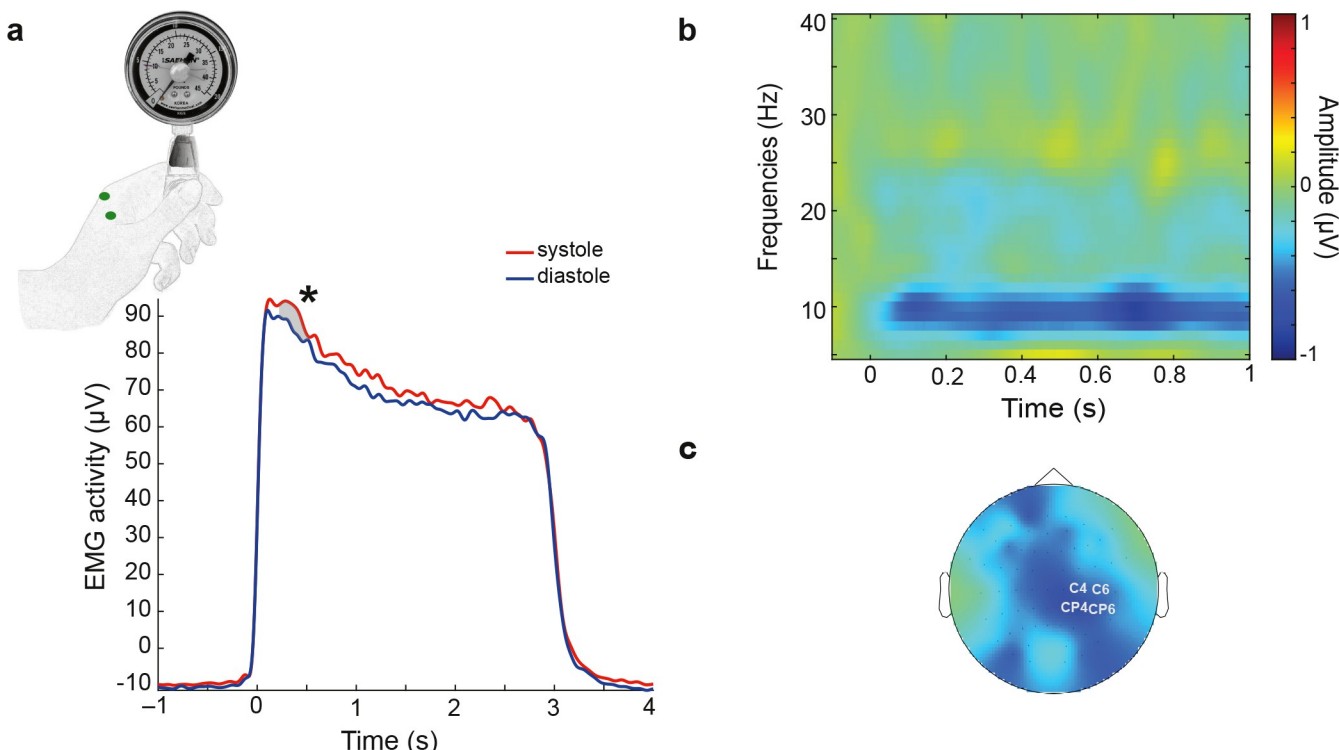

**Fig 4. Fluctuations of muscle-related activity depending on the pinch onset across the cardiac cycle during a motor pinch task.** (**a**) Muscle force, measured by the linear envelope of EMG activity in the left hand, was significantly higher from 340 to 454 ms following pinch onset (at 0 ms) during systole compared to diastole. (**b**) Similarly, systolic and diastolic sensorimotor oscillations were analyzed in the range of 8–30 Hz in the sensorimotor electrodes to quantify event-related desynchronization following the muscle activation. Cluster statistics revealed that when participants started the pinch during systole, the desynchronization of sensorimotor oscillations was higher in the frequency range of 8–25 Hz between 0 and 726 ms following pinch onset. The raster plot shows the contrast between systole and diastole. (**c**) The topography of the significant contrast between systole and diastole.

that there was no significant difference in heart rate before TMS stimulation between systole and diastole ($t_{35}$ = 1.83, $p$ = 0.075), whereas the heart rate difference was significant during TMS stimulation ($t_{35}$ = 2.10, $p$ = 0.043, Cohen's d = 0.35). This difference was no longer statistically significant in the post-TMS window ($t_{35}$ = 1.80, $p$ = 0.080).

To control whether these heart rate changes were due to genuine effects of TMS rather than artifacts (for instance, auditory, somatosensory) induced by TMS application, heart rate across time was subtracted with the heart rate during the sham TMS condition individually for each time interval and cardiac phase. This analysis again showed a similar main effect of time ($F_{2, 70}$ = 7.42, $p$ = $1\cdot10^{-3}$, $ges$ = $4\cdot10^{-3}$) and an interaction of time and cardiac phase ($F_{1.56, 54.64}$ = 3.88, $p$ = 0.03, $ges$ = $3\cdot10^{-3}$) on heart rate. These results suggest that heart rate is influenced by TMS only when administered during systole but not during diastole.

## Heartbeat-evoked potentials fluctuate depending on motor excitability levels

The analysis of cardiac phase effects on motor excitability showed a contrasting pattern to the previous findings on somatosensory perception. Here, we asked whether a similar contrast would be observed for another aspect of heart–brain interactions, which is the cortical responses to heartbeats, so-called heartbeat-evoked potentials (HEPs). Therefore, we analyzed the relationship between HEPs and motor excitability levels. To be able to distinguish cardiac and TMS-related neural processing, we first only chose trials in which the TMS stimulation

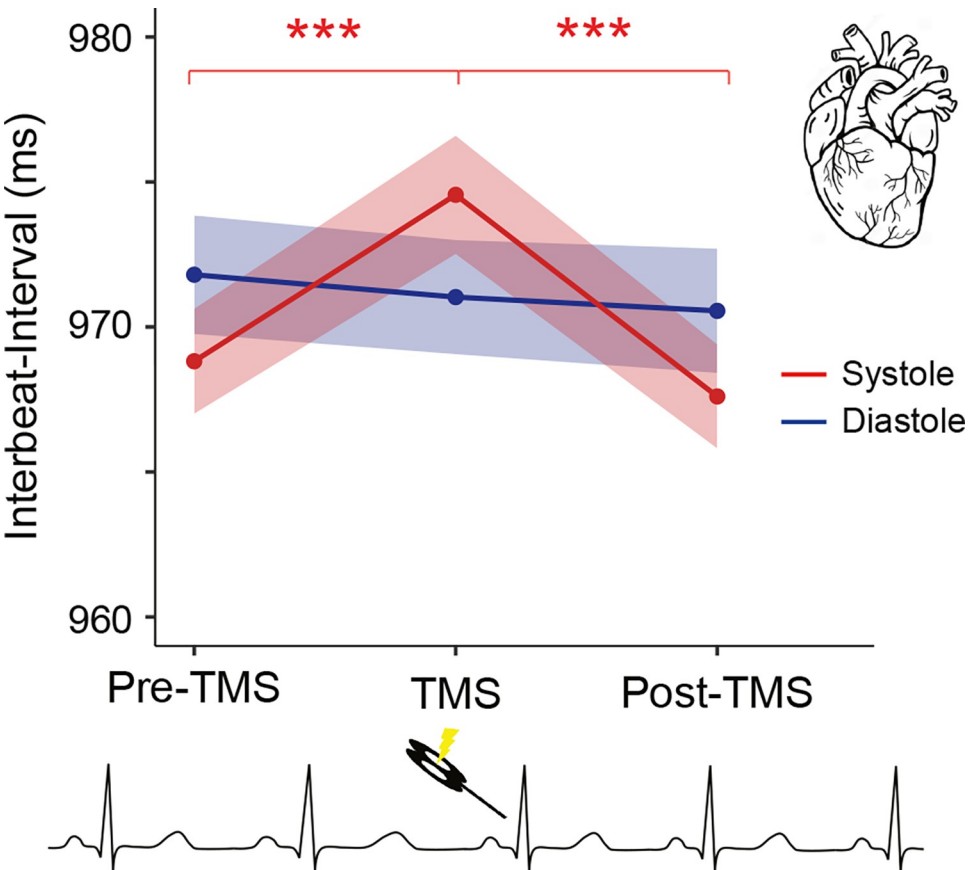

**Fig 5. Heart rate changes induced by TMS are influenced by the cardiac cycle.** The heart first slowed down and then accelerated when TMS pulses were delivered during systole. No significant differences were observed for stimulations during diastole. Colored bands indicate 95% within-participant confidence intervals. ***$p < 0.0005$.

occurred at least 400 ms after the preceding R-peak in line with the previous studies [11,13]. This allowed us to investigate the relationship of HEPs (during systole) with MEPs (during diastole) within the same cardiac cycle. For this purpose, we sorted individual trials based on their MEP amplitudes, ranging from weak to strong, and divided them into 3 equal bins for each participant. The sorting was based on MEP amplitudes since they were present in every single trial included in the analysis. Subsequently, we compared the prestimulus HEP amplitudes preceding weak (first bin) and strong (third bin) MEP levels. This comparison was conducted using a cluster-based permutation $t$ test within the time window of 296 to 400 ms post R-peak, focusing on the centroparietal electrodes as identified in previous studies [11,13]. HEPs were significantly higher preceding strong compared to weak MEP amplitudes, between 304 and 324 ms over the centroparietal electrodes ($p_{cluster} = 0.021$ corrected for multiple comparisons in space and time; Cohen's d = 0.48; **Fig 6A and 6B**). We then asked whether MEP amplitudes during the first 400 ms of the cardiac cycle are influenced by HEP activity in response to the previous cardiac cycle. This analysis showed that HEP amplitudes were more positive between 362 and 394 ms at centroparietal electrodes preceding strong compared to weak MEP amplitudes ($p_{cluster} = 0.018$ corrected for multiple comparisons in space and time; Cohen's d = 0.53; **S1 Fig**).

To test whether these effects are induced by overall changes in cardiac activity during TMS stimulation, we further tested the differences in HEP activity during TMS stimulation and

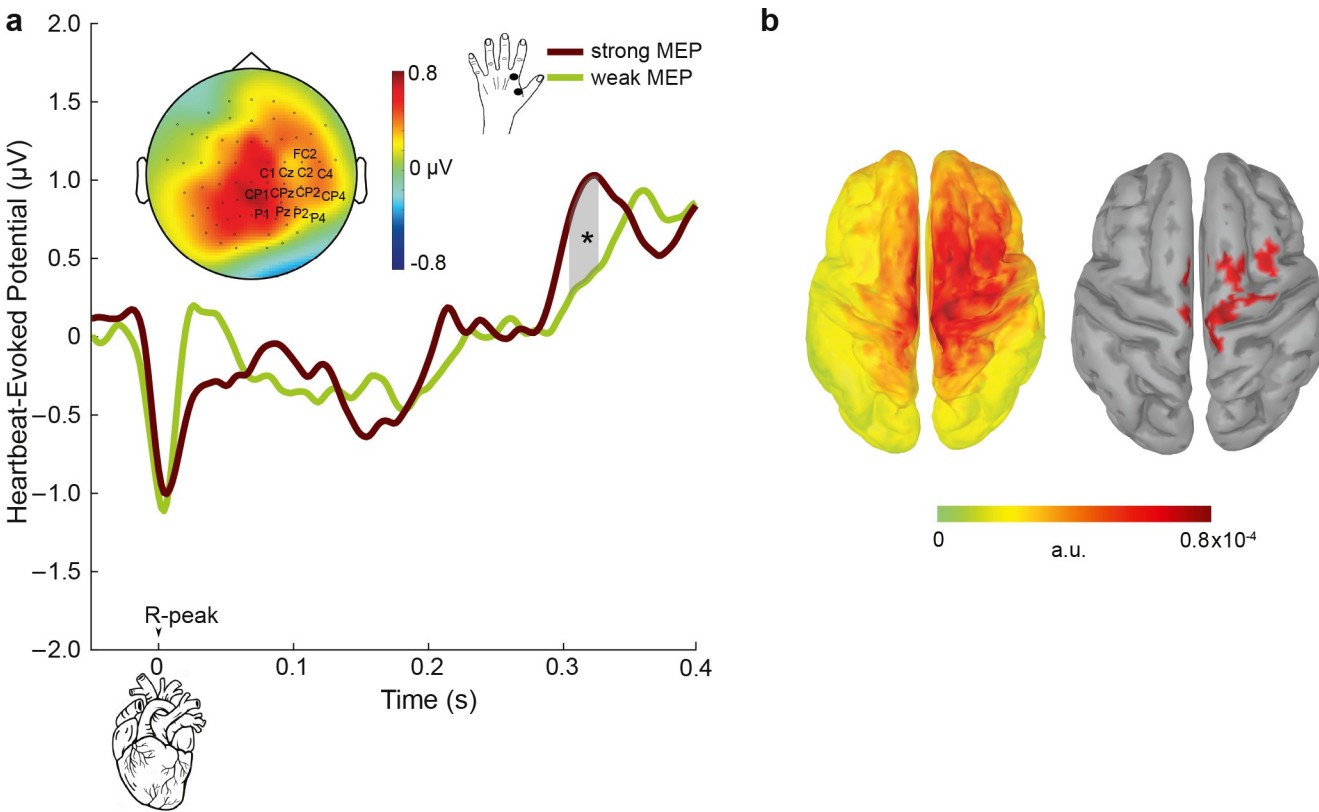

**Fig 6. The cortical responses to heartbeats, HEPs, preceding changes in the strength of motor excitability.** (**a**) To assess relationship between HEPs and motor excitability, single trials were sorted according to MEP amplitudes and split into 3 equal bins for each participant. HEP amplitudes between 304 and 328 ms following the R-peak (the highlighted gray area) were higher preceding strong compared to weak MEPs across the centroparietal electrodes. The topography contrast represents HEP amplitude difference preceding strong versus weak MEPs between 304 and 328 ms. (**b**) (left) The neural sources of HEP differences preceding strong and weak MEPs are visualized. (right) Same as the left figure but displaying the strongest generators only (thresholded at 85% of the maximum activity and clusters sizes of at least 5 vertices). *$p < 0.05$, ***$p < 0.0005$.

resting-state condition (without TMS stimulation). No significant differences in HEP amplitudes during TMS and resting-state were observed (no clusters were found; S2 Fig). Overall, these analyses show that increases in HEP amplitudes lead to stronger motor excitability levels, which again stands in contrast to the previous HEP findings in somatosensory perception.

## Control analyses for movement of cortex across the cardiac cycle

Next, we investigated whether our findings of larger motor excitability during systole might have been related to the displacement of the cortex due to blood hemodynamics (as a result of cardiac activity). Previous research has shown that the mechanical displacement of the cortex follows an inverse u-shaped pattern across the cardiac cycle and reaching its maximum at 450 to 500 ms after the R-peak [33]. As the electric field induced by a TMS pulse is stronger when the coil is closer to the brain [34], any heart phase–dependent modulation of MEPs could potentially be influenced by cortical tissue movement towards the skull. If this was the case, we would expect a stronger TMS stimulation at the peak of the displacement. Therefore, we examined whether our findings displayed a similar temporal pattern as observed in the displacement of cortical tissue to shed light on this aspect. To investigate whether MEP amplitudes followed a similar pattern across the cardiac cycle, we first visualized MEP amplitudes across 50 ms bins following the previous R-peak (S3 Fig). This demonstrated that MEPs were

maximal during the first 50 ms and gradually decreased across the cardiac cycle. To statistically test the relationship between MEP amplitudes and the distance from the previous heartbeat, linear-mixed-effects model regressions were fit on the single-trial level. The linear regression that included the cardiac distance (MEP ~ distance + (1 | participant)) explained the empirical data better than the null model, i.e., a model with no relationship assumed (MEP ~ (1 | participant); $\chi^2$ = 4.67, $p$ = 0.03). Crucially, this linear model also showed a better fit than a second-degree polynomial model (MEP ~ distance + distance^2+ (1 | participant), $\chi^2$ = 0.34, $p$ = 0.6). This result indicates that the changes in MEP amplitudes do not follow a u-shaped pattern across the cardiac cycle and are thus not likely to be explained by the displacement of the cortex due to blood influx or efflux.

## Discussion

Using simultaneous recordings of brain activity, heart activity, and muscle activity, this study discovered that the timing of heartbeats and their neural processing were linked to changes in the excitability of the motor system. Our findings build upon previous evidence showing that somatosensory perception is affected during the systolic phase of the cardiac cycle and when heartbeats evoke stronger cortical responses. To delve deeper into these cardiac effects, we examined whether they arise from overall changes in cortical excitability. For this, we performed simultaneous recordings of cortical and corticospinal excitability using electroencephalography (EEG) and electromyography (EMG) responses to TMS, along with concurrent monitoring of cardiac activity using electrocardiography (ECG). As a result, we observed multiple complementary effects of cardiac signals on motor activity. Specifically, we discovered that cortical and corticospinal excitability reached their highest levels during systole and were enhanced following stronger neural responses to heartbeats. Additionally, in a motor task, we observed that hand–muscle activity and the associated desynchronization of sensorimotor oscillations were more pronounced during systole. These findings suggest that systolic cardiac signals exert a facilitatory effect on motor excitability, which contrasts with the sensory attenuation previously reported in somatosensory perception. Therefore, our study sheds light on the existence of distinct time windows across the cardiac cycle that potentially optimize perception and action.

### Importance of methodology for current findings

Methodological differences are likely to explain the absence of cardiac modulation of motor excitability in previous TMS studies [26–28]. In those studies, time resolution across the cardiac cycle was rather limited. Unlike in our study, TMS stimulations were not presented throughout the entire cardiac cycle, they were rather presented up to 400 or 600 ms after the heartbeats and at specific time points (for instance, 100 ms after R-peak). Since our study revealed a linear decrease in motor excitability throughout the cardiac cycle, previous studies might not have been able to sample the decrease towards the end of the cardiac cycle. Furthermore, combining a neuronavigational system with individual brain scans provided us with a higher spatial specificity for the stimulation location, in comparison to previous studies. In addition, our study had the advantage of a larger sample size and trial numbers, which contributed to higher statistical power to detect the cardiac-cycle effects on motor excitability [26–28]. Since the observed MEP effect was rather small in our study, a larger set of participants (as in the present study) would indeed facilitate a statistical detection of differences. Finally, previous studies only tested cardiac effects on peripheral MEPs without concurrent cortical recordings. Yet, it is known that peripheral MEPs reflect excitability changes at both cortical and spinal cord levels and special measures should be taken (such as H-reflex) in order to

disentangle excitability changes between these 2 levels [35]. On the other hand, TEPs directly reflect changes in cortical excitability, especially at early latencies [36]. This can also explain why we observed a stronger effect size for the cardiac modulation of TEPs as compared to MEPs.

## Central and peripheral changes of motor excitability across the cardiac cycle

Cardiac cycle effects on motor excitability are consistent with previous findings of increased frequency of muscle movement during systole as compared to diastole [18,21,23,24]. Here, we showed that TMS during systole is associated with higher corticospinal and cortical excitability in motor areas. Thus, motor-related activity seems to be facilitated during systole. This, in turn, may also explain why eye movements [23,24], for instance, (micro)saccades, and voluntary hand movements [21], for instance, firing a gun, have been found to occur more often during systole. One could also argue that the effects we found were merely due to the fact that the distance of the brain to the skull (and, thus, the TMS coil) changes due to fluctuations in intracranial pressure throughout the cardiac cycle [37]. This, in turn, should affect the induced electric field from TMS. However, our control analyses did not support this argument. More specifically, the movement of the brain follows an inverse u-shaped pattern across the cardiac cycle and reaches a maximal distance at about 450 to 500 ms after the previous heartbeat [33]. If the cortical movement across the cardiac cycle was responsible for the cardiac phase effects, then MEPs would be expected to follow a similar pattern across the cardiac cycle. However, MEP amplitudes decreased rather linearly across the cardiac cycle, differently than effects that would be expected due to cortical movement. This result indicates that the changes in MEP amplitudes are not likely to be explained by the displacement of the cortex across the cardiac cycle. Another possible artifact, which can influence the amplitude of the evoked activity, are muscle-related far-fields from the cardiac activity (typically referred to as "cardiac artifacts" in the EEG). To control for those, we included a sham condition, in which auditory, tactile, and cardiac artifacts were comparable. After the correction of real TMS recordings with the sham condition, cortical excitability was still significantly higher during systole as compared to diastole. Overall, we found that the excitability in the sensorimotor cortex is increased in systole and correspondingly such increase is mirrored in the strength of the motor output. However, the strength of the motor output is not the only parameter important for motor control. The timing of the movement initiation as well as the coordination between different muscle groups could be equally important for the functionally relevant motor output. These aspects should be investigated in future studies.

The cardiac cycle was also observed to affect muscle activity in a motor task, where participants were asked to pinch and release a dynamometer with their left index finger and thumb. When the pinch was initiated during systole, compared to diastole, muscle activity was transiently stronger, suggesting a systolic increase in the applied force [38]. In addition to the peripheral activity, we analyzed cardiac effects on the central neural activity during the motor task. Previous studies have shown that following muscle contractions, sensorimotor oscillations desynchronize in the motor regions, which is reflected as an amplitude decrease in the alpha and beta range [39,40]. Here, we found that this desynchronization transiently increased when the pinch was initiated during systole. Furthermore, these cardiac effects on the muscle-related activity are not likely due to cardiac artifacts, since no significant differences in muscle and neural activity were observed across the cardiac cycle while participants were resting. Overall, these findings suggest that muscle activity is stronger when movement is initiated during systole due to an increase in motor excitability.

## Underlying mechanisms and the link to perception

The increased motor excitability during systole seems to be at odds with the previously shown cardiac effects on perception. For example, we recently demonstrated that somatosensory percepts and their neural processing are attenuated during systole [11,13]. To compensate for this reduced perceptual sensitivity, a recent study revealed that touches initiated in the systole phase were held for longer periods of time [17]. These findings could be explained by an interoceptive predictive coding account, which postulates that rhythmic cardiac signals are predicted and suppressed from entering conscious perception. This mechanism was suggested to additionally inhibit the perception of coincident weak external stimuli [11,13]. Furthermore, this suppression of nonsalient sensory stimuli was suggested to lead to a greater uncertainty about threatening factors in the environment [41]. To compensate for it, the organism might increase expectation for a "risk" and use its limited resources for a "flight or fight" motor response, which can be potentially mediated by increased baroreceptor activity during systole. Therefore, it is possible that the increased motor activity during systole might provide a survival advantage. Hence, this would suggest that there are different optimal windows for action and perception throughout the cardiac cycle. This idea also fits well with previous studies on "sensory gating," in which somatosensory perception and evoked potentials were shown to be attenuated during movement [42–44]. This would also explain why eye movements occur more often during systole, whereas eyes fixate on visual stimuli more often during diastole [23]. Given that action has an inhibitory effect on perception, it is plausible that systolic facilitation of action is indeed consistent with inhibition of perception during the systolic phase of the cardiac cycle.

It is also important to note that different cardiac effects occur within the systolic and diastolic cardiac phases. During systole, in response to changes in blood pressure, baroreceptors become maximally active approximately 300 ms after R-peak [45]. This change in baroreceptor activity has been suggested to be the driving force of the cardiac effects on perception. For example, during this time window, the processing of somatosensory and pain stimuli is shown to be minimal [11,13,46]. These findings were in line with the idea that activation of baroreceptors cause an overall decrease in cortical excitability [14,47,48]. However, in the current study, we observed that motor excitability was strongest during the pre-ejection period of the systole, before arterial pressure starts to increase and activate baroreceptors. Therefore, the cardiac-related effects on the motor domain might not be mediated primarily by baroreceptor activity but rather via a direct neural pathway involving cardiac afferent neurons. These neurons receive inputs from sensory receptors within the heart and transmit signals to coordinate heart rate and other cardiac functions. They fire at specific phases of the ECG signal, such as around the R or T-waves, depending on their location and transduction characteristics. Importantly, they have a fast conduction velocity leading to a rapid cortical activation [49]. Future studies could investigate this idea by using animal models.

## Possible clinical relevance of heart rate changes following TMS stimulation

Cortical excitability changes have been associated previously with epilepsy, disorders of consciousness, stroke, and depression [50,51]. To counterbalance these abnormalities in cortical excitability, therapeutic applications of TMS have been introduced, for instance, for treating depression [51] or facilitating motor recovery during neurorehabilitation in stroke [50]. Our results on cardiac modulations of cortical excitability raise some important questions for these clinical populations using different TMS protocols. For example, it remains unknown how cortical excitability over the cardiac cycle is modulated in those pathological conditions. Furthermore, our observation that TMS induces changes of the heart rate during systole, i.e., when the cortical processing of heartbeats occurs, but not during diastole, can have important

implications for the clinical use of TMS. For instance, in case the changes in heart rate during TMS application are of clinical concern for patients, our results, based on single pulse TMS, suggest that stimulation during diastole may help mitigate these undesired effects. Addressing these questions in future research could provide valuable insights for patients undergoing TMS in clinical settings.

### Relationship between heartbeat-evoked potentials and motor excitability levels

Another effect of cardiac activity on motor excitability was found on the cortical level. We observed that HEPs, during systole, showed higher positivity over centroparietal electrodes between 304 and 328 ms preceding strong as compared to weak corticospinal excitability (as measured by TMS-induced MEPs). These results again diverge from our previous results on somatosensory perception, in which we observed higher HEPs preceding attenuated somato-sensory processing. We previously explained increases in HEPs as a result of an attentional switch from the external world to internal bodily signals, such as heartbeats [11]. This was further supported by higher HEP amplitudes when participants were resting compared to engaging in an external task [13]. If internal attention levels changed in the current study during the TMS condition compared to rest, we would expect lower HEPs during the TMS condition. However, in the current study, there was no significant change in HEPs during the TMS application in comparison to the resting state of the participants. This was probably related to the absence of an external task during the TMS condition. Another factor, which can positively influence HEP amplitude, is arousal [52]. Increases in arousal are also known to increase motor excitability [53] as well as heart rate [1,54]. Supporting a possible involvement of arousal in our study, we observed that heart rate became higher as motor excitability increased. Therefore, we suggest that increases in arousal might be responsible for increases in HEP amplitudes for stronger motor excitability. It is also possible that since this analysis involves HEPs and MEPs, which were close in time, there was a similar cortical state for both responses due to intrinsic neuronal dynamics. If the magnitude of both HEPs and MEPs reflects increased cortical excitability, then a positive correlation would be expected, since cortical excitability changes on many time scales [55], including a period covering both pre- and immediate post-stimulus intervals.

### Limitations

While the present study provides valuable insights into the influence of cardiac oscillations on motor excitability, there are several limitations that should be acknowledged. Firstly, the study sample consisted of young white individuals between the ages of 18 and 40. Although we antici-pate that our findings may extend to the broader population, it is crucial for future studies to investigate the effects of cardiac oscillations in more diverse participant populations across different racial groups. Secondly, we only stimulated the right motor cortex. Further research should examine whether similar cardiac effects on motor excitability can be observed for left-hemispheric stimulation. Finally, we did not examine the potential impact of respiratory oscil-lations on the observed results. As respiration has been shown to organize neural dynamics [56] and modulate cortical excitability [57], it would be important for future studies to investi-gate how respiratory phase (i.e., inhale and exhale) during TMS stimulation and movement initiation might influence the effects of cardiac oscillations.

### Concluding remarks

In conclusion, our study provides novel insights into the regulation of cortical and corticospi-nal excitability by cardiac function in healthy individuals. Together, these findings strongly

suggest that systolic cardiac activity and its cortical processing have facilitatory effects on motor excitability, in contrast to the previous findings on somatosensory perception. Thus, we propose that optimal windows for action and perception are likely to differ across the cardiac cycle. Furthermore, these results may contribute to the development of novel stimulation protocols and promote a better understanding of the interplay between brain dynamics and bodily states in both health and disease.

## Methods

### Participants

Based on our previous study in which we observed an effect of the cardiac cycle on somatosensory perception [12], we performed a power analysis (using the "pwr" package in R). Given a Cohen's d of 0.48, we calculated the required sample size as $N = 36$ (80% power and alpha = 0.05) for observing cardiac effects of this size.

For the experiment, we only invited participants who were between 18 and 40 years old and who did not report any neurological, cognitive, or cardiac health problems. Exclusion criteria further included tinnitus, alcohol or drug abuse, and pregnancy. A total of 37 healthy volunteers participated in the experiments after giving written informed consent. All protocols were approved by the Ethical Committee of the University of Leipzig's Medical Faculty (Ethics no: 179/19) and followed the principles of Declaration of Helsinki. One participant was excluded due to failure in the data acquisition. In the remaining 36 participants (20 female, age: 27.97 ± 4.13, mean ± SD), only one participant was left-handed as assessed using the Edinburgh Handedness Inventory [58].

### TMS setup and neuronavigation

The experiment included 4 blocks of sham and 4 blocks of real TMS stimulations. Participants were seated in a comfortable armchair and asked to keep their eyes on a fixation point on a wall in front of them throughout the measurements. TMS pulses were delivered through a Magstim 200 Bistim stimulator (Magstim, Whitland, United Kingdom) connected to a figure-of-eight coil (Magstim "D70 Alpha Coil"). The coil was positioned at an angle of 45° with respect to the sagittal direction. Structural T1 weighted MRIs of the participants were used with the TMS neuronavigation system (Localite GmbH, Bonn, Germany) to identify the hotspot of the left first dorsal interosseous (FDI) muscle. Then, the resting motor threshold was determined as the lowest TMS intensity at which 5 out of 10 trials yielded a motor response greater than 50 μV (peak-to-peak amplitude) [59]. The TMS blocks consisted of 104 trials, i.e., a total of $104 \times 8 = 832$ stimulations. The neuronavigation system was used to control the coil position over the hotspot during the TMS stimulations. TMS intensity was set to be 20% above the motor threshold at rest (corresponding to 66.58 ± 9.16% of the maximum stimulator output). The interstimulus interval was uniformly randomized between 1.5 and 2.5 s. The blocks were presented as pairs of 2 sham or real TMS, and their order were randomized across participants. For the sham TMS condition, we used a custom-manufactured 3.5 cm plastic block between the coil and the participant's head to keep air- and bone-conducted auditory sensations similar to the real TMS [31]. This setup also mimicked a tapping somatosensory sensation associated with the vibration of the TMS coil.

### EEG, ECG, and EMG recordings

TMS-compatible EEG equipment (NeurOne Tesla, Bittium) was used for recording EEG activity from the scalp. The EEG was acquired with a bandwidth of 0.16 to 1250 Hz from 62 TMS-

compatible c-shaped Ag/AgCl electrodes (EasyCap GmbH, Herrsching, Germany) mounted on an elastic cap and positioned according to the 10–10 International System. POz electrode was used as ground. During the measurements, the EEG signal was referenced to an electrode placed on the left mastoid. Additionally, a right-mastoid electrode was recorded so that EEG data could be re-referenced to the average of both mastoid electrodes offline. The signal was digitized at a sampling rate of 5 kHz. Skin/electrode impedance was maintained below 5 kΩ. EEG electrode positions were also coregistered with the structural MRIs using the neuronavigation system. To reduce auditory response artifacts in the EEG induced by coil clicks, participants wore earplugs throughout the experiment. An additional ECG electrode connected to the EEG system was placed under the participant's left breast to record the heart activity. Furthermore, EMG electrodes were attached to the left FDI muscle in belly–tendon montage via a bipolar channel connected to the EEG system to record the TMS-induced MEPs. At the beginning of the experiment, EEG and ECG data were acquired during a 5-min eyes-open resting-state measurement.

## Automated cardiac phase classification

The fluctuations of motor excitability were tested across the systolic and diastolic phases of the cardiac activity. Systole was defined as the time between the R-peak and the end of the t-wave, which was determined by using a trapezoid area algorithm [11,60]. We then used the duration of systole to define an equal length of diastole at the end of each cardiac cycle [11]. By using time windows of equal length for systole and diastole, we equated the probability of a stimulation/event occurring in either of the 2 phases. As a result, the average systole (and diastole) length was $351 \pm 21$ ms. Before using this automated algorithm, we removed large TMS artifacts on the ECG data by removing −2 to 10 ms window around the TMS stimulation and then applied cubic interpolation. As a result, the number of trials was not significantly different ($t35 = −1.05$, $p = 0.3$) between systolic ($148 \pm 17$) and diastolic ($150 \pm 18$) parts of cardiac cycle. Therefore, with this approach, we could ensure comparable trial numbers across conditions.

## Motor-evoked potentials

As an index of corticospinal excitability, MEPs were used. After applying a baseline correction using −110 to −10 ms prestimulus window, peak-to-peak MEP amplitudes were calculated in the EMG data in the time window of 20 to 40 ms following the TMS stimulation. A successful TMS stimulation is expected to trigger an MEP higher than 50 μV, whereas a successful sham application does not trigger motor-evoked activity, while having other features of stimulations such as auditory click and bone vibration [59]. We applied these criteria to ensure that only valid TMS and sham trials were included in the analysis. This yielded on average 412 TMS trials and 405 sham trials per participant.

## TMS-evoked EEG potentials

EEG data were first segmented between −1,400 and 1,000 ms around TMS stimulations. Then, the baseline correction was performed using −110 to −10 ms prestimulus window. The large-amplitude TMS artifacts between −2 and 8 ms were removed from each trial and then the remaining data segments were concatenated, in line with a previously established procedure [61]. Next, ICA (round 1) was applied using pop_runica as implemented in EEGLAB, used with the FASTICA algorithm implementing the "tanh" contrast function and a symmetrical approach [61,62]. To remove TMS decay artifacts, the 3 components explaining the most variance between −150 and 150 ms were removed and other components were forward projected [61]. After the decay artifact was removed in this way, copies of these datasets were kept. Then,

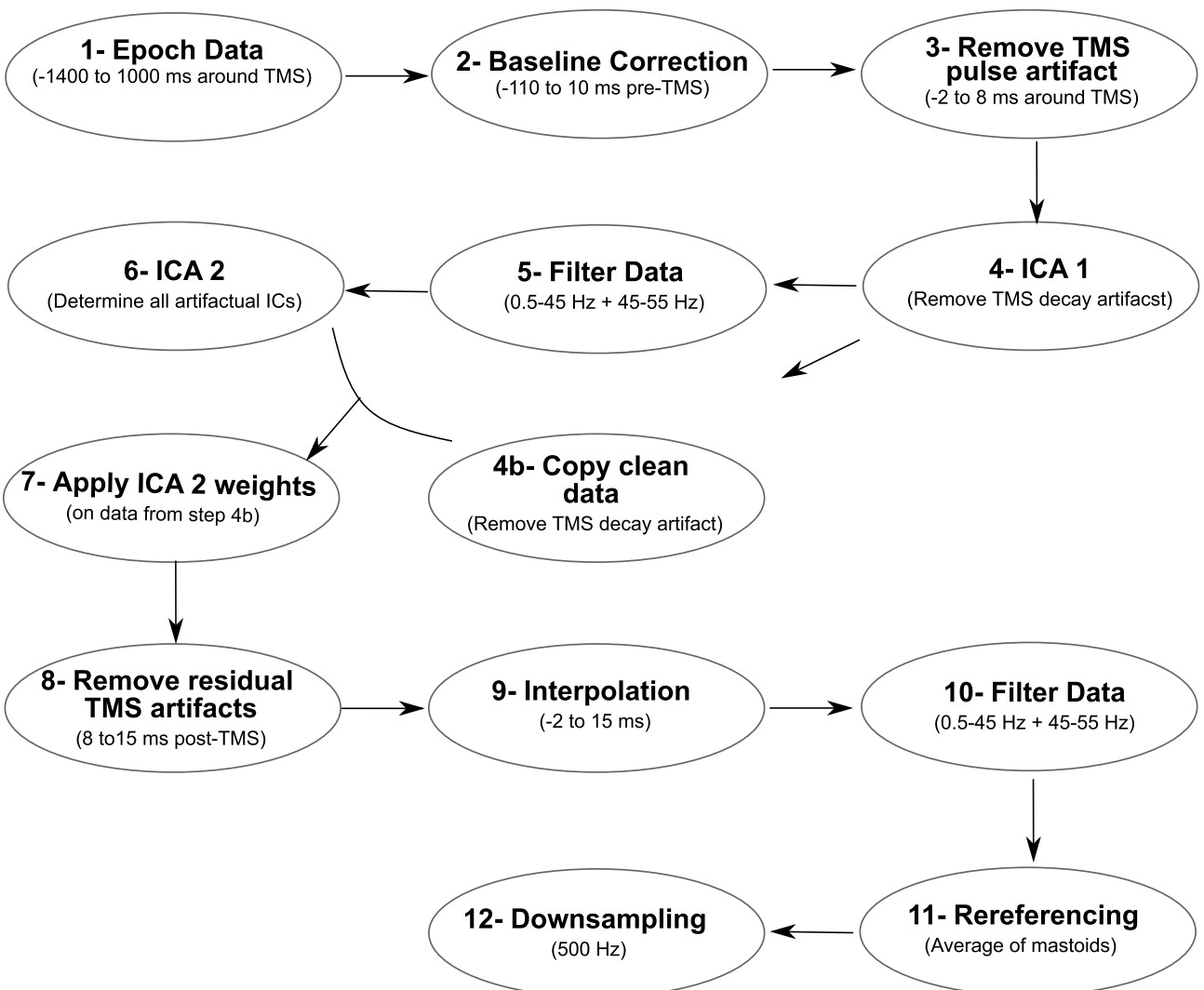

**Fig 7. Preprocessing steps of EEG data for cleaning artifacts in the post-TMS window.** These steps were followed before calculating TMS-evoked potentials (TEPs).

a fourth-order Butterworth bandpass filter (0.5 to 45 Hz) and a 50-Hz notch filter (with a stop-band of 45 to 55 Hz) were applied. A second round of ICA was applied to determine remaining TMS, ocular, muscle, and cardiac artifacts [61]. Afterwards, these ICA weights were applied to the copied dataset after the first round of ICA (unfiltered). On average, 11 ± 2 artifactual components were removed and then the data were forward projected. Due to remaining TMS-evoked artifacts, we further removed 15 ms poststimulus window. We then applied a cubic interpolation (from −2 to 15 ms relative to TMS stimulation) before applying the same filtering procedure (as described above) to the data [61]. The interpolated data were only used for data visualization and excluded from statistical analyses. This way we ensured that the TMS artifacts did not smear into the poststimulus window during the filtering process. Then, data were re-referenced offline to the average of the right and left mastoid signals and down-sampled to 500 Hz (**Fig 7**).

To assess cortical excitability in motor areas, we focused on early components of TEPs in the first 60 ms following the stimulation, since this time window specifically involves the

activation of local neuronal populations in the motor cortex [36]. Since the first 15 ms were interpolated, we evaluated TEPs between 15 and 60 ms in the post-TMS window in a cluster of electrodes over the right primary motor cortex (C4, CP4, C6, CP6). These electrodes were selected a priori since they were closest to the hotspot of the TMS stimulation.

Cardiac artifact during systole and diastole was estimated during TMS and sham conditions (see [11] for details on the pulse artifact cleaning of the evoked potentials) and subtracted from TEPs during systole and diastole individually.

### Heartbeat-evoked potentials

In this analysis, we first analyzed the trials in which TMS stimulation was at least 400 ms after the previous R-peak (i.e., during diastole) to keep the HEP window free of TMS-related activity following previous studies [11,13]. This allowed us to examine the relationship of HEPs (in response to heartbeats during systole) with MEPs (in response to TMS stimulation during diastole) within the same cardiac cycle. In this study, we also tested whether MEP amplitudes during the first 400 ms of the cardiac cycle are influenced by HEP activity from the previous cardiac cycle.

To clean TMS artifacts in the prestimulus window and keep data processing close to our previous work [11,13], some preprocessing steps were altered compared to the steps described above for the poststimulation analyses: After the second round of ICA, we first calculated each distance between the prestimulus R peaks and TMS events. Then, we shuffled these distances and inserted "mock events" by subtracting them from the latency of TMS stimulations in the dataset. Next, we repeated this shuffling process 10 times. Finally, we segmented data between −100 and 400 ms around these mock events. By using an average of these segments, we derived an estimate of the TMS artifact in the time window of the heartbeat-evoked responses per participant.

We then subtracted this estimation from each HEP to remove any potential TMS artifacts. Finally, a baseline removal was performed, using the time window from −100 to 0 ms (relative to the R-peak).

### Source analyses

The neural sources of the TEPs and HEPs were reconstructed with the Brainstorm toolbox [63] using individually measured electrode positions with a TMS neuronavigation system (Localite GmbH, Bonn, Germany). For every participant, the individual structural T1-weighted MRI images were segmented using Freesurfer (http://surfer.nmr.mgh.harvard.edu/). A 3-shell boundary element model (BEM) was constructed to calculate the lead field matrix with Open-MEEG [64,65]. The lead field matrices were inverted using eLORETA individually for each contrast (for instance, TEP difference between systole and diastole) and participant. Individual source data were then projected to the ICBM152 template [66]. Following this for each participant, the absolute values for each contrast were calculated. These absolute values were then used to compute grand averages of the HEP and TEP.

### Motor pinch task

After the TMS sessions, participants performed a pinch motor task. At the beginning of the task, their maximal pinch strength, i.e., maximal voluntary contraction (MVC), was calculated using SAEHAN Hydraulic Hand Dynamometer Model SH5005 (SAEHAN Corporation, Korea). Participants were asked to squeeze the dynamometer with their left thumb pad against the lateral aspect of the middle phalanx of the left index finger as hard as possible while keeping their elbow in the 90˚ position. After calculating MVC, participants first practiced applying

30% of MVC across several trials consistently. During the task, participants were asked to apply the 30% of MVC (corresponding to 3.14 ± 0.88 pounds) when a green circle in the middle of the monitor returned to red. During the presentation of the red circle, which lasted for 3 s, they were asked to keep the contraction. When the circle became green again, they relaxed their hand for 3 s. In this order, participants performed contractions for 30 trials.

### Movement onset estimation

The onset of movement was determined by analyzing EMG activity. For this purpose, we used a Matlab package, EMG-onset-detection. This package uses a method based on Teager–Kaiser Energy (TKE) operator and corrections of false onset detection and EMG artifacts [67]. If automatic detection did not work, the onset of movement was determined manually. Then, we determined the corresponding cardiac phase of each movement onset. On average, each cardiac phase included 12 trials.

### EMG envelope

To estimate muscle activity during the pinch task, EMG data were analyzed. First, EMG signal was filtered by using a fourth-order Butterworth bandpass filter (10 to 500 Hz) and a 50-Hz notch filter (with a stopband of 45 to 55 Hz). Afterwards, the envelope of EMG was calculated by first taking the absolute value of the signal ("full-wave rectification") and then applying an 8-Hz low-pass filter [68]. The resulting EMG envelope was epoched between −1 and 4 s around the movement onset. This was followed by a baseline correction using the −110 to −10 ms premovement EMG signal. Finally, the envelope was averaged for trials where movement was initiated during systole and diastole per participant.

### Desynchronization of sensorimotor oscillations during the motor task

To investigate the central sensorimotor oscillations following pinch onset during systole and diastole, we also analyzed EEG signals. For this purpose, we first filtered the data with a fourth-order Butterworth bandpass (0.5 to 45 Hz) and a 50-Hz notch filter (with a stopband of 45 to 55 Hz). After cleaning muscular, cardiac, and ocular artifacts through ICA and re-referencing data to the average of both mastoid electrodes, data were segmented between −1,000 and 4,000 ms around the pinch onset. We then performed a Morlet wavelet analysis to investigate sensorimotor alpha and beta activity locked to pinch onset. This analysis was performed on every trial for frequencies from 5 to 40 Hz with the number of cycles increasing linearly from 4 to 10. Thus, a wavelet at 10 Hz was 4.9 cycles long, had a temporal resolution of 0.10 s and a spectral resolution of 4.85 Hz. We then calculated the average time-frequency activity for each cardiac phase per participant.

### Statistics

We statistically tested the two-condition comparisons of TEPs, HEPs, EMG linear envelope, and sensorimotor oscillations using cluster-based permutation $t$ tests as implemented in the FieldTrip toolbox [69]. To define clusters, the default threshold value ($p < 0.05$, two-tailed) was used. To test cluster-level statistics, condition labels were randomly shuffled 1,000 times to estimate the distribution of maximum cluster-level statistics obtained by chance. We report the temporal windows and spatial regions that we tested for each individual analysis below.

Statistical analysis of TEP activity during systole and diastole were conducted at electrodes C4, CP4, C6, and CP6 between 15 and 60 ms. Pre- and post-TMS changes in heart rate for stimulation during systole and diastole were evaluated using within-participant ANOVAs

(ezANOVA function in R, v 1.3.1093; [70,71]), in which heart rate was the dependent variable and time (pre-TMS, TMS, post-TMS) as well as cardiac phase (systole, diastole) were independent variables. For statistical testing of HEP activity relating to motor excitability, we first sorted single trials according to their MEP amplitudes and split them into 3 equal bins for each participant. We specifically used 3 bins to create a strong contrast between weak and strong MEPs while providing a sufficiently high number of trials in each category. For the weakest and strongest MEP bins, we then contrasted prestimulus HEP amplitudes between 296 and 400 ms in a cluster of electrodes (FC2, Cz, C4, CP1, CP2, Pz, P4, C1, C2, CPz, CP4, P1, P2), where we previously observed significant modulations of HEP preceding somatosensory processes [11,13]. In the cluster analysis of both TEPs and HEPs, clusters were formed in the spatiotemporal domain using the a priori defined set of electrodes and temporal windows.

During the motor task, the statistical analysis focused on the first second of the muscle contraction following the pinch onset, as cardiac effects are expected to be transient and may last for one cardiac cycle only. Therefore, statistical analysis of the EMG envelope during systole and diastole were conducted in a time window from 0 to 1,000 ms. During the same time window, sensorimotor oscillations were compared in the range from 8 to 30 Hz over a set of electrodes over sensorimotor regions (C4, CP4, C6, CP6) using cluster statistics, in order to account for multiple comparisons in the temporal, spatial, and frequency domain.

### Linear-mixed-effects model

To test the relationship between MEP amplitudes and the distance from the previous heartbeat, linear-mixed-effects models were fitted on the single-trial level. First, we hierarchically compared a null model, which assumes no relationship (MEP ~ (1 | participant) to a model that assumes a linear relationship between MEP amplitudes and the distance (MEP ~ distance + (1 | participant)). Then, we compared this linear model to a second-degree polynomial model (MEP) ~ distance + distance^2+ (1 | participant). In these models, we used the natural logarithmic transformation of MEP amplitudes.

### Supporting information

**S1 Fig. The influence of previous heartbeat-evoked potentials (HEPs) on motor excitability.**
(TIF)

**S2 Fig. Heartbeat-evoked potential (HEP) during rest and TMS stimulation.**
(TIF)

**S3 Fig. MEP amplitudes across the cardiac cycle.**
(TIF)

### Author Contributions

**Conceptualization:** Esra Al, Tilman Stephani, Melina Engelhardt, Arno Villringer, Vadim V. Nikulin.

**Data curation:** Esra Al, Tilman Stephani, Vadim V. Nikulin.

**Formal analysis:** Esra Al, Tilman Stephani, Arno Villringer, Vadim V. Nikulin.

**Funding acquisition:** Esra Al, Arno Villringer, Vadim V. Nikulin.

**Investigation:** Esra Al, Arno Villringer, Vadim V. Nikulin.

**Methodology:** Esra Al, Tilman Stephani, Melina Engelhardt, Arno Villringer, Vadim V. Nikulin.

**Project administration:** Esra Al, Arno Villringer, Vadim V. Nikulin.

**Resources:** Esra Al, Arno Villringer, Vadim V. Nikulin.

**Software:** Esra Al.

**Supervision:** Arno Villringer, Vadim V. Nikulin.

**Validation:** Esra Al, Vadim V. Nikulin.

**Visualization:** Esra Al.

**Writing – original draft:** Esra Al.

**Writing – review & editing:** Esra Al, Tilman Stephani, Saskia Haegens, Arno Villringer, Vadim V. Nikulin.

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
