## [Editor Report · Decision Letter 0]

2 Mar 2023

Dear Dr Al, 

Thank you for submitting your manuscript entitled "Cardiac Activity Impacts Cortical Motor Excitability" for consideration as a Research Article by PLOS Biology.

Your manuscript has now been evaluated by the PLOS Biology editorial staff as well as by an academic editor with relevant expertise and I am writing to let you know that we would like to send your submission out for external peer review.

Once your full submission is complete, your paper will undergo a series of checks in preparation for peer review. After your manuscript has passed the checks it will be sent out for review. To provide the metadata for your submission, please Login to Editorial Manager (https://www.editorialmanager.com/pbiology) within two working days, i.e. by Mar 06 2023 11:59PM.

Kind regards,

Luke

Lucas Smith, Ph.D.

Associate Editor

PLOS Biology

lsmith@plos.org

---

## [Decision Letter · Decision Letter 1]

27 Apr 2023

Dear Dr Al,

Thank you for your patience while your manuscript "Cardiac Activity Impacts Cortical Motor Excitability" was peer-reviewed at PLOS Biology. It has now been evaluated by the PLOS Biology editors, an Academic Editor with relevant expertise, and by several independent reviewers.

In light of the reviews, which you will find at the end of this email, we would like to invite you to revise the work to thoroughly address the reviewers' reports.

As you will see below, the reviewers comment that manuscript offers potentially interesting new insights and that the study is generally well done. However they have each raised a number of important points that will need to be addressed before we can consider your manuscript for publication. We think that an theme that emerges from the reviews, and which will require particular attention, is that the writing of the manuscript needs to be revised to improve the coherence of the story. In particular, we suggest the results be carefully rewritten to ensure that each set of analyses is preceded by a motivation (what was the aim? why was it done?) and followed by a short conclusion (what exactly did we learn from this analysis?) as we think this will improve the broad accessibility of the manuscript.

In addition to addressing the reviewer comments, we also have a number of editorial reporting and policy related requests which we ask that you attend to. I have appended those below the reviews. 

Given the extent of revision needed, we cannot make a decision about publication until we have seen the revised manuscript and your response to the reviewers' comments. Your revised manuscript may be sent for further evaluation by all or a subset of the reviewers.

**IMPORTANT - SUBMITTING YOUR REVISION**

*Re-submission Checklist*

*Published Peer Review*

Sincerely,

Lucas

Lucas Smith, Ph.D.

Associate Editor

PLOS Biology

lsmith@plos.org

REVIEWS:

Reviewer #1: Al et al., examined whether cortical and corticospinal excitability changes as a function of the cardiac phases (systole and diastole). They examined such a potential modulation via TMS (to measure excitability), with EEG to measure TMS and heartbeat-evoked potentials, and ECG to obtain the corresponding phase-locked measures. The authors found higher excitability and greater TMS-evoked potential in the systole phase of the cardiac cycle (vs. sham condition). In addition, they also recorded EMG during a simple motor task (pinching) and examined muscle-related activity. The subsequent results indicate that EMG activity was higher circa 220-520ms after the pinch onset. If the pinch was initiated in systole, desynchronization of sensorimotor oscillations was higher in the frequency range of alpha and beta oscillations. The authors conclude that the systole phase of the cardiac cycle may be related to motor facilitation (action) whereas the diastole phase to perception. 

I read the preprint a while ago, and overall, the authors have done an excellent job. The study involves an amalgam of techniques that is not usually found elsewhere. The manuscript addresses pertinent issues in the literature and moves forward the various fields (e.g., interoception). Overall, the authors have produced a stimulating piece of work. The following minor comments, suggestions, and clarifications may enhance the manuscript's scope and clarity. Please find below the comments by section:

Introduction and discussion: 

In the introduction, the authors refer to the study by Ohl et al., which demonstrated that microsaccades are coupled to the cardiac cycle. However, the way the manuscript is currently written, it seems as though two studies (references 17-18) showed this finding. In contrast, study 18 (Galvez-Pol et al.) revealed that saccades are locked around systole and visual fixations in diastole, without presenting microsaccade data. Relatedly, Galvez-Pol et al. also suggested that the diastole phase of the cardiac cycle may be linked to sensory gain (perception), which should be more accurately referenced, as the idea behind the current manuscript has been previously noted.

Furthermore, a recent paper (https://doi.org/10.7554/eLife.78126) showed that human subjects freely moved their index finger to tap a key during systole, a finding that is consistent with active visual tasks without perceptual discrimination requirements (see Kunzendorf et al., 2019). The same study discovered that when participants had to sense gratings, if touch initiation occurred during systole, they extended the duration of the touch, whereas during diastole, touches were shorter. This implies that participants needed more time for perception when it began in systole or sought to reach a more sensitive phase of the cycle. This work also supports the current manuscript; however, the introduction and discussion do not sufficiently address it. By incorporating these points, the current manuscript will be better situated within the existing literature.

Results:

My primary concern is the clarity of the rationale underpinning the analyses and the subsequent results. In its current state, the manuscript features numerous analyses with insufficient or unclear reasoning. Enhancing this aspect should not be challenging: considering the journal's broad readership, I recommend providing a more comprehensive explanation of the rationale behind the analyses that led to the results. This can be achieved by including the thought process and bits of explanations at the beginning of each subsection, particularly in the second half of the results section (e.g., IBIs and HEPs). By doing so, the manuscript's framework and the coherence of the narrative that the authors aim to present can be enhanced.

Relatedly, it is not clear to me the rationale behind the IBIs section. Why the TMS applied to the motor area should affect distinctively the duration of heartbeats as a function of cardiac phase? 

Figure 1: State briefly in the text accompanying the figure that the diastole window has been equated to the length of the systole one for analyses purposes. 

MEPs and figure 2: I suggest the authors to plot the scatter data of the MEPs with the box plots (i.e., individual dots depicting the means or medians of the subjects). If the plot changes the resolution of the axis and the difference between conditions is not easily discernible, the authors can always add a new plot of the difference by subtracting the systole and diastole MEPs (as visual aid). 

Figure 4: If I understand correctly, event-related desynchronization following muscle activation was computed for the frequency range between 8 and 25Hz. Were alpha and beat analysed separately or together? The analysis seems to indicate 'together' while the topoplots 'separate'. If the latter is true, I advise consistency between analysis and figures; show one topoplot instead of two. 

Discussion:

Regarding the studies about shooting in systole: also note that elite rifle shooters pull the trigger during diastole, with beginners firing during either phase but with better results during diastole (Helin et al., 1987). It has been suggested that this effect is related to the mechanical movement caused by the heart's contraction (Konttinen et al., 2003).

In the discussion section, I advise the authors to write a section of Constraints on Generality (COG); see Simmons et al., 2017. This involves adding some additional information about the constraints of the current work in other scenarios, tasks, paradigms, populations, etc. (see examples in the Simmons' paper). This should not be too long. This section could be also combined with the second paragraph of the discussion (which is more centred on methodological features of the present and past studies). 

The current results can be mentioned within recent discoveries on the relationship between breathing and neural function to provide a more comprehensive understanding of the interactions between physiological processes and cognition. By incorporating (briefly) the findings on the respiratory cycle, the scope of the paper can be broadened, and the potential interactions between cardiac and respiratory cycles in modulating neural activity can be discussed. For instance, the research of Kluger et al., (2021) and Allen's review on the topic (2022) can be mentioned in this context.

Methods:

To enhance clarity, what is the reason for the interpolation between -2 and 15ms in step 9? (Aligning the data, missing data..)

Independent Component Analysis (ICA) is a widely used technique for removing artifacts in EEG data. A good point is that It separates the mixed signals into statistically independent components, which can then be analyzed and classified as either neural or artifactual. I believe the implementation of ICA benefits from the data filtering (step 5) in Figure 7. How were the parameters chosen for this filter? (I thought 1Hz was recommended). Also, can the authors provide additional information about the removed ICA? (e.g., did the authors follow any rule or visual inspection? And/or how many ICs were approximately removed in average?). One issue with this sort of study is the quantity of removed data from the recording. In this context, researchers using ICA and removing TMS or cardiac artifacts may face ambiguity in component classification. In other words, identifying and classifying independent components as artifacts or neural signals can be challenging and may require expert knowledge. This can lead to false positives (removing neural signals) or false negatives (retaining artifacts). Also, ICA assumes statistical independence. ICA relies on the assumption that sources are statistically independent. If this assumption is violated, the separation of artifacts from neural signals may be less accurate.

Reviewer #2: The manuscript describe an interesting and timely study on the modulation of cortical and cortico-spinal excitability by cardiac cycle phases (systole vs. diastole). The authors test two alternative hypotheses underlying the changes in perceptual threshold and motor execution, based on earlier literature that have shown both attenuation of somatosensory perception and facilitation of motor activity during systole. Their findings support facilitatory effect on motor excitability, as they show higher central and peripheral TMS-evoked responses during systole. They also report several related evidence including effects in cardiac evoked activity and heart rate, as well as stronger muscle activity and sensorimotor cortex desynchronization during systole. The paper is generally well written, and the experiments are conducted well, with carefully planned control conditions. The number of subjects is not very big, but of needed size to allow appropriate statistical testing, with appropriate power analysis (however, see comment below related to the statistics). There are some questions, suggestions and concerns I think would need to be addressed to improve the readability and confirm the significance of the study for the field. Currently the paper is not written in a generally accessible way, and therefore it is hard to say to what extent it gives outsanding contribution. The structure and multitude of the findings are presented in rather fragmented manner and do not develop into coherent picture of a) what is the current gap (and novelty the paper brings) in the literature and b) how / why are all the different experimental manipulations and measurements needed to fill this gap. 

First, it is not entirely clear to me why are the authors forming two alternative hypotheses (e.g. Intro end of page 3), one based on earlier results on *perceptual* threshold and the other based on earlier results on *motor* execution. Their research examines only motor cortex excitability and peripheral motor evoked potentials, and thus it seems more suitable to make hypothesis based on earlier studies on the motor side. Other cardiac-related effects are of course interesting and worth reporting, but I wonder whether they should be presented as equally relevant with the motor studies. Also in the discussion the impression is given that the initial assumption would be to see similar cardiac-brain coupling for perception and action; in my understanding this is not what the earlier literature would propose - I may be mistaken but clearer and more elaborate reasoning on the background would better justify the novelty of the paper. 

The research question is clear, but the way the different recordings, manipulations and analysis pipelines are used to answer to this question is not explained well. This gives a rather fragmented impression, for example in the list of results (different sub-titles) in the Result section, as well as beginning of discussion. Now it appears almost like the analysis has been proceeding on ad-hoc manner from one result to the next. Perhaps formulating better in the introduction the way in which (and why) each data/manipulation/analysis is planned to be used would clarify the significance of the list of results. For example, it took some time to figure out how cardiac phase-related changes in TEP's should be interpreted together with heart-evoked responses, where the phase information is lost (and all the responses are similarly phase-locked to heartbeat) - or with effects in heart rate. In general, the paper appears like not fully 'matured' to a form where its scientific significance would be easily grasped. This is especially problematic from the perspective of 'non-expert' readers (or readers from other fields).

The introduction is very concise, and although it reviews the relevant literature related to cardiac-brain coupling, especially the means to measure cortical excitability is introduced in rather superficial and one-sided manner. There are some recent critiques (or at least needs for caution) presented for the straightforward interpretation of TEP's as an index of cortical excitability (especially for the early time-windows, see eg. Hernandez-Pavon et al., 2023). It would provide a more solid basis for the used methodology if the characteristics and possible reservations in using TEPs as indicators of cortical excitability would be good to at least mention in the introduction. This way also the following description of the control conditions in the result section would make more sense for the reader.

Related to the above, Result section does not give enough info about the SHAM condition (how/why was it done) for integrating it to the general picture. 

Figure 1 gives helpful info about the methods for the reader to understand the results, but it could be better organized to help grasping the different parts of the experiment (cf. 3rd paragraph above). Moreover, it could be more informative if the timelines of the EEG, ECG and EMG would be synchronized (to understand their temporal relationships) and timeline should be added. A combined or synchronized timeline would also help to estimate the overlap of artefacts/responses. 

What is the reason for presenting the correlational analysis of MEP and TEP amplitude differences across cardiac cycle first as 'initial analysis' and the following analysis of the cardiac changes in MEP against the modulation of earlier TEP's as 'exploratory'. When reading the results, it is not clear what is the difference between these analysis approaches and/or what is the particular research questions they respond to. 

As a detail (and a topic outside of my expertise): the blood-circulation -related contribution to the cardiac cycle effect on pinching-related EMG envelope is controlled by the analysis of rest condition (?) - is there a possibility that this does not properly represent the situation during muscle activity, and would this influence to the interpretation? In other words, can it be that the blood-circulation related modulation is different (more prominent) when muscles are activated, and thus the contrasting of diastole vs. systole during rest is not a valid control condition to rule out this impact?

Results related to the heart rate changes; some more details should be given for interpreting the results; from what time-window was the heart rate calculated (and what measure was used? Heart acceleration or interval between beats? Where is the analysis described for the heart rate related measures? Authors controlled for the possible artefacts (auditory, somatosensory) on the effects on heart rate by adjusting the heart rate with the one during sham TMS condition; why was this procedure chosen and not conducting the similar analysis for the TMS condition (i.e. time x diastole/systole)? Or is this what authors mean by adjusting?

Results regarding the HEP & MEP analysis; what was the way of categorizing trials to strong vs. weak MEP (the cut point would be useful to mention in the results). 

The results (particularly for heart rate changes depending on the timing of TMS and HEP fluctuation depending on motor excitability levels) describe a rather complicated scenario with several dependencies between the TMS pulse, cortical responses and motor evoked potential - it would be really helpful for the reader to get some illustration to support the following of the analysis/results. Along the same lines, it would also guide the reading (and help to confirm the logic of the analysis) if the introduction would give at least a rough outline of the planned examination (this is mentioned also earlier). Now the result section progresses in somewhat unpredictable (but apparently meaningful) steps; for example categorizing the separate sections based on the specific part of the research question they respond to would help to keep track of the results (together with the illustration). 

Last result section, regarding the control analysis, do I understand correctly that this analysis is based on the known motion related to cardiac phases or was the analysis done based on the current data? 

I am not fully following the reasoning why MEP would be expected to follow a similar pattern as the movement of the brain, if the TMS-effects were due to the movement?

The beginning of discussion gives a nice overview on the results, but here again it may not be self-evident for the reader why and how the motor pinch task-related cardiac-related changes in muscle activity and cortical descynronization and especially the TMS-triggered heart-rate changes are linked to the original excitability changes. In my view, too much expert knowledge is expected form the reader to understand the value and meaning of the findings for the field. What is the phenomenon on which this paper brings the outstanding contribution? Why is it interesting and to what scientific debate or discussion does it contribute to? Occasionally the paper stays at the level of 'technically neat and comprehensive list of results'.

Although not specifically my own field, I wonder how the cardiac cycle influences to the ability of (peripheral) muscles to contract, and how would this influence to the interpretation of the current results?

In Discussion p. 16 the authors present nice reasoning on how the cardiac modulation of perception vs. action could be linked to each other. Do I understand correctly that the authors suggest that the driving force for both the effects in perception and motor activity arise from the need to dampen the rhythmic cardiac signals not to enter conscious perception? This explanation, building on the idea of 'noise cancellation', would naturally explain why there would be any need for rhythmic variation in the perceptual thresholds and motor activity with such a short time-scale. What if this assumption is not correct? Are there alternative ways to explain the effects and how does this reasoning link with the current literature and other speculations on the heart-brain coupling? For better linking with the current literature, it would be important to present the current findings & suggested hypotheses in the broader literature on the topic - the body-brain coupling is a very active field at the moment and there are many recent papers and reviews on the topic.

Authors also speculate their findings could reflect a direct neural pathway via cardiac afferent neurons. This is an interesting suggestion for further studies, but it contrasts with the very narrow background on anatomical (and animal model) research presented in the introduction. What is currently known about the cardiac-brain connections? Are there studies supporting direct connections? 

Methods section gives detailed protocols for each data / analysis type, but it remains unclear which part of these analysis steps are according to golden standard principles/established (and published) protocols. Whenever possible, it would help if the authors cited any available papers describing the pipelines or principles that are followed in the analysis. 

As a detail, in p. 22, line 565 onwards, authors say that only trials in which TMS stimulation triggered high-enough motor-evoked potential were included, and for sham only trials in which no evoked motor activity was observed were included. Why was this the case? It would help (especially the non-expert reader) to remind where were these categories/contrasts used in the analysis. 

In p. 23, Source Analysis, line 614, authors say that individual source data were projected to the template brain, and their absolute values were used in group averages - what group averages are referred to? 

Several different statistical tests are used for various time-windows, contrasts etc. How is the problem of multiple testing (also across different data types/contrasts) dealt with? The power analysis is only reported for limited tests, but given the numerous different data and manipulations, it may not be sufficient to guarantee rigorous statistical approach. In its current form, it is also difficult to see to what extent is the analysis based on clear hypothesis and to what extent is it more of exploratory nature.

Refs. 

Hernandez-Pavon JC, Veniero D, Bergmann TO, Belardinelli P, Bortoletto M, Casarotto S, Casula EP, Farzan F, Fecchio M, Julkunen P, Kallioniemi E, Lioumis P, Metsomaa J, Miniussi C, Mutanen TP, Rocchi L, Rogasch NC, Shafi MM, Siebner HR, Thut G, Zrenner C, Ziemann U, Ilmoniemi RJ. TMS combined with EEG: Recommendations and open issues for data collection and analysis. Brain Stimul. 2023 Feb 23;16(2):567-593. doi: 10.1016/j.brs.2023.02.009. Epub ahead of print. PMID: 36828303.

Reviewer #3, Florent Lebon (Note, reviewer 3 has signed this review): In the current study, the authors investigated the influence of cardiac cycles (systole and diastole) on corticospinal and cortical excitability and on muscle activity during contraction. They also tested the effect of transcranial magnetic stimulation on cardiac measures and the relationship between heartbeat-effect cortical activity and corticospinal excitability. Contrary to the literature, the authors found greater corticospinal and cortical activity, as well as greater muscle activity during a specific time-window of pinch movement, during systole in comparison to diastole. Overall, this study brings further information regarding the relationship between cardiac activity and cortical and corticospinal excitability. However, it does not provide evidence of functional effects.

Please find general remarks and minor comments below.

In the abstract, the authors wrote that "Cortical and corticospinal excitability were found to be highest (…) following stronger cortical responses to heartbeats". This sentence is ambiguous as cortical activity is mentioned twice (as the source and the consequence).

In the introduction section, the authors stipulated that "What remains unknown are the underlying mechanisms of these effects". The authors could be more specific to highlight the goal and the originality of their study, to explain the opposite results.

It is not clear if the authors synchronized the TMS pulses and the cardiac phases or if they stimulated, then categorized a posteriori in which cycle the TMS pulse was triggered.

l.172 "significant increase in the normalized EMG envelope". Normalized to what?

Did the authors use the force produced as a covariable? As it may have influenced the EMG signal.

l.180 "we also tested whether sensorimotor oscillations (in the range of 8 - 30 Hz)" "the desynchronization of sensorimotor oscillations in the range of 8 - 25 Hz was stronger". Why this range? And why is the range different (8-30; 8-25)?

l.234: "we sorted single trials according to their MEP amplitudes and split them into three equal bins for each participant". Why did the authors choose 3 bins (and not 2 or 4)?

From l. 203 and l.255: it is not clear why the authors investigated "the changes in the heart rate in response to TMS stimulation" and "the effect of motor excitability on heart rates". After focusing on the modulations of central and peripheral changes across the cardiac cycles, the authors analyzed the other way around. They should clarify why these analyses were important to answer the general problematic. 

L.260 "as motor excitability increased, the heartbeats became faster (cardiac acceleration)". Could we also say that "as heartbeats became faster, motor excitability increased"? How can the authors interpret the causality between heartbeat and MEP amplitude modulation? And I would say corticospinal or corticomotor (not just motor) excitability when interpreting MEP amplitude modulation.

l.366 "Overall, these results suggest that motor excitability is higher during systole, suggesting an optimal window for motor activity across the cardiac cycle". Could the authors be more specific about this argument? Do the small (but significant) differences in cortical and corticospinal excitability across the cardiac cycles prevail for functional efficiency?

l.468: "Given a Cohen's d of 0.48, we calculated the required sample size as N=36 (80% power and alpha=0.05) for observing cardiac effects on perception". Why do the authors refer to 'perception' here?

l.627: "The trials were categorized into systole and diastole according to the movement onset of the subject after the visual cue". Was the beginning of the contraction synchronized with the cardiac cycles? Or was the categorization performed a posteriori? How many trials during systole?

Also, between the presentation of the Go signal, and the beginning of the contraction, there is a reaction time (between 130 and 150ms if the instructions are to move as fast as possible at the imperative signal). How did the authors make sure that the analyzed trial was categorized into the right category? Did they take into account the presentation of the signal or the beginning of EMG activation? EMG activation could begin during systole while the presentation of the signal was during diastole, and inversely.

l.638: "an average of the envelope was calculated when pinch onset coincided with the systolic and diastolic phases of the cardiac cycle per subject". What was the window for averaging EMG envelope?

EDITORIAL REQUESTS: 

1) TITLE: After some discussion, we think the title could be strengthened to more directly reflect the findings of your study. If you agree (and if supported), we might suggest it be changed to something like "Systolic cardiac signals facilitate motor excitability'

2) FINANCIAL DISCLOSURES: I noticed in your financial disclosure that you report having received no specific funding for this work. Even if you did not have a grant specifically written to fund this project, we would require that you acknowledge the sources of money that funded this project, including salaries, etc. You should also acknowledge institutional funding or private grants as well, if relevant.

3) ETHICS STATEMENT: Please update your ethics statement to note whether your study was conducted according to the principles expressed in the Declaration of Helsinki.

4) BLURB: When re-submitting, please provide a blurb which (if accepted) will be included in our weekly and monthly Electronic Table of Contents, sent out to readers of PLOS Biology, and may be used to promote your article in social media. The blurb should be about 30-40 words long and is subject to editorial changes. It should, without exaggeration, entice people to read your manuscript. It should not be redundant with the title and should not contain acronyms or abbreviations.

5) DATA AVAILABILITY: You may be aware of the PLOS Data Policy, which requires that all data be made available without restriction: http://journals.plos.org/plosbiology/s/data-availability. For more information, please also see this editorial: http://dx.doi.org/10.1371/journal.pbio.1001797

Note that we do not require all raw data (we understand that your protocol does not permit the public distribution of the raw data collected in your study). Rather, we ask that all individual quantitative observations that underlie the data summarized in the figures and results of your paper be made available in one of the following forms:

a - Supplementary files (e.g., excel). Please ensure that all data files are uploaded as 'Supporting Information' and are invariably referred to (in the manuscript, figure legends, and the Description field when uploading your files) using the following format verbatim: S1 Data, S2 Data, etc. Multiple panels of a single or even several figures can be included as multiple sheets in one excel file that is saved using exactly the following convention: S1_Data.xlsx (using an underscore).

b- Deposition in a publicly available repository. Please also provide the accession code or a reviewer link so that we may view your data before publication. 

Fig 2B; Fig 3; Fig 4; Fig 5; Fig 6; Fig S1; Fig S2; Fig S3;

---Please also ensure that figure legends in your manuscript include information on where the underlying data can be found, and ensure your supplemental data file/s has a legend.

---Please ensure that your Data Statement in the submission system accurately describes where your data can be found.

---

## [Decision Letter · Decision Letter 2]

28 Sep 2023

Dear Dr Al,

Thank you for your patience while we considered your revised manuscript "Cardiac Activity Impacts Cortical Motor Excitability" for consideration as a Research Article at PLOS Biology. Your revised study has now been evaluated by the PLOS Biology editors, the Academic Editor and the original reviewers, who are largely satisfied by the changes made in the revision, but who have a number of additional suggestions and concerns that we think should be addressed. 

In light of the reviews, which you will find at the end of this email, we are pleased to invite an additional, short revision to address the remaining points from the reviewers. We will then assess your revised manuscript and your response to the reviewers' comments with our Academic Editor aiming to avoid further rounds of peer-review, although might need to consult with the reviewers, depending on the nature of the revisions.

**EDITORIAL REQUESTS: Thank you also for addressing our previous editorial requests in this revision. For the most part, we are satisfied by your responses, but we noticed a couple of lingering issues with the response to our data policy requests. 

1) We see that you have provided code and subject data as a deposition to OSF. Thank you for this. Can you please update the OSF link to include a readme file detailing what is contained there and how it relates to the figures?

2) We understand that you are not able to share the raw EEG and MRI data related to your study due to consent issues. This is OK, under our acceptable data restriction policy (https://journals.plos.org/plosone/s/data-availability#loc-acceptable-data-access-restrictions) - but we ask that you please update your data availability statement to provide contact information for a data access committee, ethics committee, or other institutional body to which data requests may be sent (rather than directing researchers to a corresponding author). This provides researchers with a more durable point of contact to ensure data will be accessible even if an author changes email addresses, institutions, or becomes unavailable to answer requests.

3) While we understand that you cannot provide the raw data related to your study, I wonder if for some figures you can provide the anonymized summary statistics that underlie the graphs presented here? For example, can you provide, in an excel or on OSF, the individual quantitative observations that underlie the data summarized in the figures 5, S1, S3?

**IMPORTANT - SUBMITTING YOUR REVISION**

*Resubmission Checklist*

*Published Peer Review*

*PLOS Data Policy*

*Blot and Gel Data Policy*

Sincerely,

Lucas

Lucas Smith, Ph.D.

Senior Editor

PLOS Biology

lsmith@plos.org

REVIEWS:

Reviewer #1: Al et al. investigated the relationship between cardiac phases (systole and diastole) and cortical as well as corticospinal excitability through an amalgam of techniques: TMS, EEG, and ECG. Elevated excitability was observed during the systole phase, alongside increased EMG activity during a simplistic motor task. These findings potentially indicate a correlation between the systole phase and motor facilitation. Upon reviewing both the preprint and subsequent versions, it is apparent that the authors have judiciously employed diverse methodologies to address salient questions, particularly in the realm of interoception. Overall, the manuscript has been substantially improved, aligning well with prior reviewer feedback. The revisions have enhanced the paper's scope, clarity, and primary conclusions. Minor observations, which are readily addressable, are as follows:

General Comments:

1) The results indicating increased excitability are modest, with a p-value approximating 0.040. Consequently, I recommend circumspect language in both the abstract and discussion where these findings are discussed. For example, in the abstract, the phrase "Cortical and corticospinal excitability were found to be highest during systole" could be cautiously reworded. Additionally, consideration should be given to the use of words like "may" in statements like "distinct time windows 'may' exist across the cardiac cycle that either optimize perception or action.". Overall please revise the sentences along the manuscript where absolute/superlative terminology are used to state the results of excitability. Also, noting at least once that such results yielded modest significance would give a transparent and clear image of the present work. 

2) The manuscript's discussion section, similar to the introduction and results in the earlier version, suffers from a disordered flow of ideas. Adding subheadings to delineate different content types (e.g., Cortical Excitability and Cardiac Cycle, Brain Activity in Systole and Diastole, Further Research and Limitations, etc.) may enhance readability and engagement. Otherwise, the discussion is sometimes read as a bit of chaotic amalgam of ideas/results.

Specific Comment:

The study by Galvez-Pol et al. (2020) mentioned that pulsations due to blood circulation could contribute to observed effects of the cardiac cycle on cognition but did not delve/research this aspect. When referencing this idea in the text, a more nuanced statement or even deleting such statement would be prudent. Otherwise it seems that they (Galvez-Pol) studied such a research question and gave a result accordingly. 

Reviewer #2: The authors have clarified the manuscript making it more accessible and the structure now better supports also the understanding of the different data types. All my questions has been thoroughly answered and when relevant manuscript modified accordingly.

As a minor puzzling point: the rationale/motivation for including the TMS induced effects in heart rate is still not clear to me based on the introduction; the added sentence only mentions it is included leaving the topic rather separate from the rest of the introduction. However, in the discussion, heart rate results are elaborated, making it clearer and more understandable how this analysis/section link with the overall scope of the study. I wonder if it would be possible to add some more reasoning from this perspective to introduction. This issue does not require re-review.

Typo: line 158-159; neuronal responses e by TMS, it should not be present in the sham condition.

Reviewer #3, Florent Lebon (note, this reviewer has signed their review): The authors responded point-by-point to my remarks.

While they made substantial modifications within the manuscript to provide a clear message, I still have few general concerns, which could be easily addressed.

General concerns

I do not know the policy of the journal, but it seems odd to provide the results at the end of the introduction section (l.82-89), without any further information about the protocol. I leave the editors decide whether the authors should keep the structure as it is.

In the current form, the result section looks like an accumulation of analyses with a lack of coherence between the different sub-sections. While the two first paragraphs ("Motor-evoked potentials change across the cardiac cycle", and "TMS-evoked potentials vary across the cardiac cycle") are obvious to respond to the main hypothesis, the other parts (e.g., "Muscle-related peripheral and central activity fluctuates across the cardiac cycle", "Heart rate changes depending on the timing of TMS across the cardiac cycle") are less likely to provide a clear message. I suggest the authors to provide a short sentence at the beginning of each result sub-section, explaining why this is important to perform these analyses. This will help the reader understand the common thread.

Is there any relationship between the paragraphs "Heart rate changes depending on the timing of TMS across the cardiac cycle" line 203 and "Heart rate fluctuates depending on motor excitability levels" line 255? Both deal with "changes in heart rate due to the cardiac timing of TMS". Is there any chance that both paragraphs could be merged? Or at least one after the other?

The paragraph lines 415-430 should be revised. In rehabilitation, repetitive TMS is used to improve the condition of the patients. Single-pulse TMS (as the method used in this study) provides information about a neurophysiological status at the time of the stimulation. Repetitive TMS includes blocks of stimulation, for which the duration is between 3 and 15 minutes. In the article by Johansen-Berg et al. (reference 45), single-pulse TMS is used to estimate the involvement of specific brain in regions in motor control comparing healthy individuals and stroke patients. I would suggest the authors to revise this perspective section, focusing on the evaluation (single or paired-pulse TMS) and not (or less) the treatment (rTMS).

Minor changes

P.2, l.48 : delete « . , » before « though »

---

## [Editor Report · Decision Letter 3]

19 Oct 2023

Dear Dr Al,

Thank you for the submission of your revised Research Article "Cardiac Activity Impacts Cortical Motor Excitability" for publication in PLOS Biology, and thank you also for addressing the last editorial and reviewer requests in this revision. On behalf of my colleagues and the Academic Editor, Joachim Gross, I am pleased to say that we can in principle accept your manuscript for publication, provided you address any remaining formatting and reporting issues. These will be detailed in an email you should receive within 2-3 business days from our colleagues in the journal operations team; no action is required from you until then. Please note that we will not be able to formally accept your manuscript and schedule it for publication until you have completed any requested changes.

PRESS

Sincerely, 

Lucas Smith, Ph.D.

Senior Editor

PLOS Biology

lsmith@plos.org